# Phertilizer: Growing a clonal tree from ultra-low coverage single-cell DNA sequencing of tumors

**Leah L. Weber**[1]☯, **Chuanyi Zhang**[2]☯, **Idoia Ochoa**[2,3]*, **Mohammed El-Kebir**[3,4]*

**1** Department of Computer Science, University of Illinois Urbana-Champaign, Urbana-Champaign, Illinois, United States of America, **2** Department of Electrical & Computer Engineering, University of Illinois Urbana-Champaign, Urbana-Champaign, Illinois, United States of America, **3** Department of Electrical and Electronics Engineering, University of Navarre, Donostia, Spain, **4** Cancer Center at Illinois, University of Illinois Urbana-Champaign, Urbana-Champaign, Illinois, United States of America

☯ These authors contributed equally to this work.
* idoia@illinois.edu (IO); melkebir@illinois.edu (MEK)

**Data Availability Statement:** Code and data are available at https://github.com/elkebir-group/phertilizer.

**Funding:** M.E-K. was supported by the National Science Foundation (CCF-2046488) as well as

## Abstract

Emerging ultra-low coverage single-cell DNA sequencing (scDNA-seq) technologies have enabled high resolution evolutionary studies of copy number aberrations (CNAs) within tumors. While these sequencing technologies are well suited for identifying CNAs due to the uniformity of sequencing coverage, the sparsity of coverage poses challenges for the study of single-nucleotide variants (SNVs). In order to maximize the utility of increasingly available ultra-low coverage scDNA-seq data and obtain a comprehensive understanding of tumor evolution, it is important to also analyze the evolution of SNVs from the same set of tumor cells. We present PHERTILIZER, a method to infer a clonal tree from ultra-low coverage scDNA-seq data of a tumor. Based on a probabilistic model, our method recursively partitions the data by identifying key evolutionary events in the history of the tumor. We demonstrate the performance of PHERTILIZER on simulated data as well as on two real datasets, finding that PHERTILIZER effectively utilizes the copy-number signal inherent in the data to more accurately uncover clonal structure and genotypes compared to previous methods.

## Author summary

The development of a tumor can be explained using a phylogeny—a tree that describes the evolutionary history and has therapeutic implications. A tumor phylogeny can be constructed from single-cell DNA sequencing data but each technology has advantages and disadvantages. In particular, ultra-low coverage technologies sequence the genome uniformly, which facilitates accurate inference of mutations that impact copy number in a genomic region within each cell. However, the sparse coverage makes it difficult to study the evolution of point mutations, which only impact a single DNA base. Consequently, there are no existing algorithms to infer the evolutionary history of point mutations from this single-cell technology. In this work, we propose PHERTILIZER, a method to infer a tumor phylogeny of point mutations from ultra-low coverage technologies that uses the

funding from the Cancer Center at Illinois. I.O. was supported by a Gipuzkoa Fellows grant from the Basque Government, a Ramon y Cajal Grant from Spain, and a grant from the Spanish Ministry of Science and Innovation (PID2021-126718OA-I00). This work used resources, services, and support provided via the Greg Gulick Honorary Research Award Opportunity supported by a gift from Amazon Web Services. The funders had no role in study design, data collection and analysis, decision to publish, or preparation of the manuscript.

**Competing interests:** The authors have declared that no competing interests exist.

strong copy-number signal to overcome the sparse coverage. Our results suggest that PHERTILIZER is able to more accurately infer the evolutionary history of point mutations in a tumor compared to existing *ad hoc* approaches. This improved accuracy can help yield new insights into patterns of tumor progression across and within different patient groups, ultimately getting us closer to improving our basic understanding of cancer and how to design and apply treatment.

## Introduction

Cancer results from an evolutionary process that yields a heterogeneous tumor composed of multiple subpopulations of cells, or *clones*, with distinct sets of somatic mutations [1] (Fig 1a). These mutations include single-nucleotide variants (SNVs) that alter a single base and copy-number aberrations (CNAs) that amplify or delete large genomic regions. Over the last decade, new developments in single-cell DNA sequencing (scDNA-seq) methods have helped uncover a wealth of insights regarding intra-tumor heterogeneity and cancer evolution [2–5]. In particular, the ongoing development and application of high-throughput, ultra-low coverage scDNA-seq technologies (<1×), such as direct library preparation (DLP+) [6] and acoustic cell tagmentation (ACT) [7], have paved the way for enriching our understanding of the role CNAs play in cancer progression and tumor evolution [6–8].

The advantage these ultra-low coverage scDNA-seq technologies have over other high-throughput scDNA-seq methods (>1×), like Mission Bio Tapestry [9], is the uniformity of

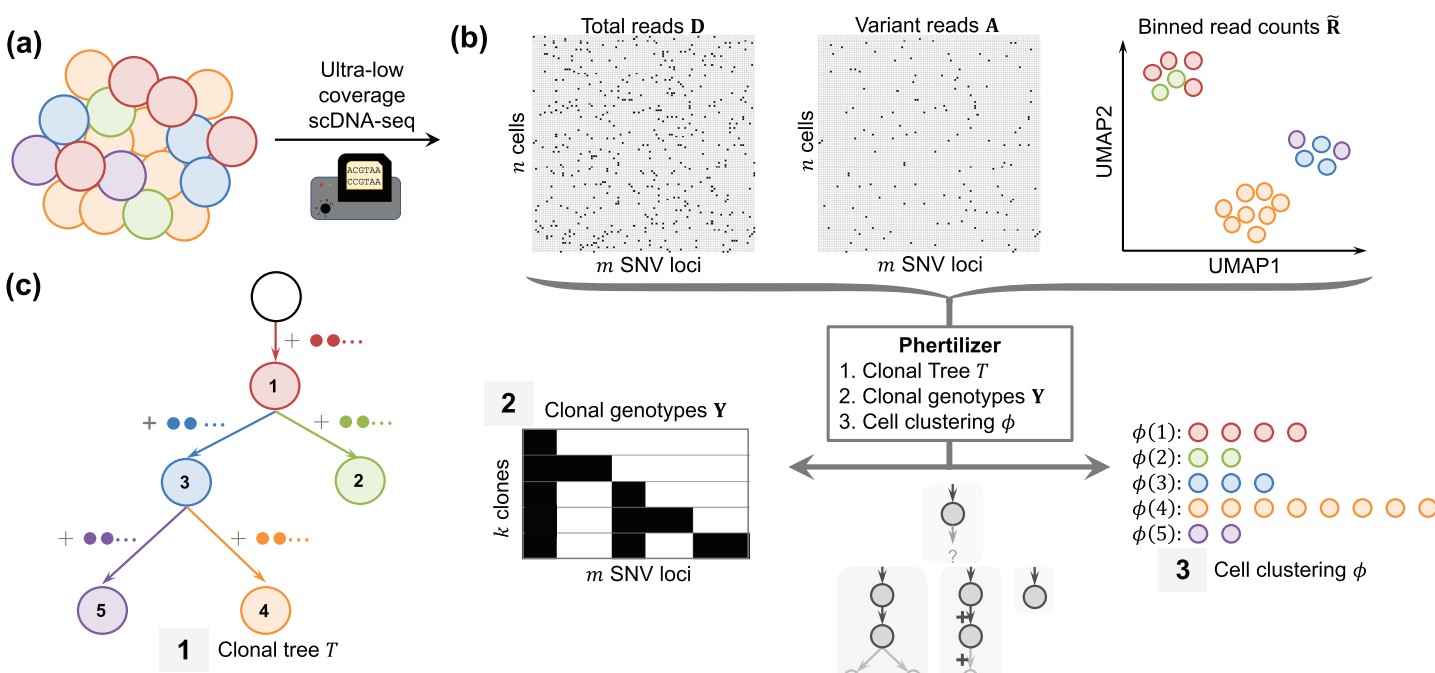

**Fig 1. PHERTILIZER infers a clonal tree $T$, clonal genotypes $\mathbf{Y}$ and a cell clustering $\phi$ given ultra-low coverage single-cell sequencing data.** (a) A tumor consists of clones with distinct genotypes. (b) Ultra-low coverage scDNA-seq produces total read counts $\mathbf{D} \in \mathbb{N}^{n \times m}$ and variant read counts $\mathbf{A} \in \mathbb{N}^{n \times m}$ for $n$ cells and $m$ SNV loci, and low dimension embedding of binned read counts $\tilde{\mathbf{R}} \in \mathbb{R}^{n \times \ell}$. (c) Given maximum copy number $c$ and sequencing error probability $\alpha$, PHERTILIZER infers a clonal tree $T$, clonal genotypes $\mathbf{Y}$ and cell clustering $\phi$ with maximum posterior probability $P(T, \mathbf{Y}, \phi \mid \mathbf{A}, \mathbf{D}, \tilde{\mathbf{R}}, c, \alpha)$.

coverage. This uniformity implies that the observed read counts for a genomic region is proportional to copy number, making it ideal for the analysis of subclonal CNAs that occur in only a small subset of tumor cells. However, this uniformity comes at the cost of sequencing depth, making it very difficult to identify and characterize the evolution of SNVs from ultra-low coverage scDNA-seq. Critically, to comprehensively study the evolution of a tumor from the same set of cells, both CNAs and SNVs should ideally be characterized by a single tumor phylogeny that depicts their coevolution. While this remains a long-term goal for the field, a first step in this direction is to increase our understanding of SNV evolution from ultra-low coverage scDNA-seq data by incorporating reliable copy number information into the inference of SNV tumor phylogenies. Although phylogeny inference methods from bulk sequencing and cell clustering from single-cell RNA sequencing are expanding to incorporate both SNV and CNA features, such as TUSV-ext [10] and CASIC [11], current methods for tumor phylogeny and/or clone inference from single-cell sequencing naturally tend to focus on the features (SNV or CNA events) for which the data is ideally suited [12–26]. One exception is BiTSC2 [24], which infers a phylogeny containing both SNV and CNA events. However, the method is not designed for high-throughput ultra-coverage data, with efficacy demonstrated on datasets containing at most 500 cells and coverages as low as 3×. Another exception in the medium to high coverage scDNA-seq regime (>10×) is SCAR-LET [27], which refines a given copy number tree using SNV read counts under a CNA loss supported evolutionary model. While SCARLET accounts for sequencing errors and missing data, it was not designed to handle the extreme sparsity of ultra-low coverage scDNA-seq. SBMCLONE [28] took the first step of using ultra-low coverage sequencing data to infer SNV clones via stochastic block modeling. Despite good performance on simulated data, especially with higher coverage (>0.2×), SBMCLONE was unable to identify clear structure in a 10x Genomics breast cancer dataset [4] without *ad hoc* use of additional copy number clone information. Moreover, it is non-trivial to convert the inferred parameters of SBMCLONE's stochastic block model to clonal genotypes, which may impact downstream analyses. Similarly, given a set of candidate SNV loci, SECEDO [29] first calls SNVs using a Bayesian filtering approach and then subsequently clusters cells using the called SNVs. While both of these clustering methods capitalize on the ever-increasing throughput of ultra-low coverage scDNA-seq methods, neither method constrains the output by a tree and CNA features are used only in an *ad hoc* manner or for orthogonal validation. As a result, both methods imply that CNA and SNV data features should be segregated and analyzed in separate bioinformatics pipelines. The other emerging trend from the analysis of ultra-low coverage scDNA-seq is pseudobulk analysis [6]. This approach begins by identifying copy number clones using existing methods, followed by pooling cells that belong to the same CNA clones into pseudobulk samples, which are then independently analyzed to identify SNVs. Finally, phylogeny inference is performed with the copy number clones as the leaves of the tree. By doing so, this method does not allow for further refinement of these clones based on SNV evolution.

Here we introduce PHERTILIZER, the first method to infer an SNV clonal tree from ultra-low single-cell DNA sequencing of tumors. To overcome SNV coverage sparsity in this type of data, we leverage the strong copy number signal inherent in the data to guide clonal tree inference. By analogy to the planting and growing of trees, PHERTILIZER seeks to grow a clonal tree with maximum posterior probability by recursively inferring elementary clonal trees as building blocks. Our simulations demonstrate that PHERTILIZER accurately infers phylogenies and cell clusters when the number of cells matches current practice. In particular, PHERTILIZER outperforms a current method [28] for simultaneously clustering SNVs and cells as well as another commonly used *ad hoc* approach [6]. On real data, we find that PHERTILIZER effectively

utilizes the copy-number signal inherent in the data to uncover clonal structure, yielding high-fidelity clonal genotypes.

## Materials and methods

### Problem statement

Our goal is to infer an SNV phylogeny, guided by copy number aberrations, from ultra-low coverage sequencing data consisting of $n$ cells and $m$ identified single-nucleotide variants (SNVs). More precisely, we are given variant reads $\mathbf{A} = [a_{iq}]$ and total reads $\mathbf{D} = [d_{iq}]$, where $a_{iq}$ and $d_{iq}$ are the variant and total read counts for SNV locus $q \in [m]$ in cell $i \in [n]$, respectively (Fig 1b). While the number of cells is large with the latest generation of ultra-low coverage single-cell DNA sequencing technology ($n \approx 1000$ cells), the *coverage*, or the average number of reads that span a single locus is uniform but low ($0.01\times$ to $0.5\times$). For example with a coverage of $0.01\times$, we would on average observe $a_{iq} = d_{iq} = 0$ reads for 99 out of every 100 loci $q$ for each cell $i$. This sparsity renders phylogeny inference using SNVs extremely challenging. We propose to overcome this challenge using the following three key ideas.

First, similarly to current methods, we leverage the clonal structure present in tumors, i.e., cells typically cluster into a small number of clones. Thus, we seek to group the $n$ observed cells into $k$ clones ($k \ll n$) via a cell clustering and corresponding clonal genotypes defined as follows:

**Definition 1** *Function $\phi$: $[k] \to 2^{[n]}$ is a cell clustering provided its image encodes a partition of cells $[n]$ into k (disjoint and non-empty) parts.*

**Definition 2** *Matrix $\mathbf{Y} \in \{0, 1\}^{k \times m}$ encodes clonal genotypes where $y_{jq} = 1$ indicates that SNV q is present in clone j and $y_{jq} = 0$ indicates that SNV q is absent in clone j.*

Second, because ultra-low coverage scDNA-seq is sequenced uniformly, we may leverage the copy number signal inherent in the data to improve cell clustering performance [11] and guide tree inference. More specifically, we expect that all cells in a clone have identical copy number profiles. As we do not observe copy number directly, we will use observed reads counts $\mathbf{R} \in \mathbb{N}^{n \times b}$, where $b$ is the number of genomic bins, as a proxy for copy number. From read counts $\mathbf{R}$, we derive distances that reflect copy-number similarity between pairs of cells on a low-dimensional embedding of binned read counts $\tilde{\mathbf{R}} \in \mathbb{R}^{n \times \ell}$ of $\mathbf{R}$ (i.e., $\ell \ll b$)— see Section A.1 in S1 Appendix. Third, similarly to methods such as SCITE [12], SciCloneFit [15] and SPhyR [14], which operate on medium-to-high coverage scDNA-seq data, we consider that the observed cells are generated as the result of a tree-like evolutionary process that constrains the order of SNV clusters. In particular, we use the infinite sites model [30] defined as follows.

**Definition 3**. *A tree T with nodes $\{v_1, \ldots, v_k\}$ rooted at node $v_1$ is a* clonal tree *for clonal genotypes $\mathbf{Y} = [\mathbf{y}_1, \ldots, \mathbf{y}_k]^\top$ provided (i) each node $v_j$ is labeled by clonal genotype $\mathbf{y}_j$ and (ii) each SNV q is gained exactly once and subsequently never lost. That is, there exists no directed edge $(v_{j'}, v_j)$ where the SNV is lost, i.e., $y_{jq} = 1$ and $y_{j'q} = 0$. Moreover, either the root node contains the SNV, i.e., $y_{1q} = 1$, or there exists exactly one directed edge $(v_{j'}, v_j)$ where the SNV is introduced, i.e., $y_{j'q} = 0$ and $y_{jq} = 1$.*

To relate the data to our latent variables of interest $(T, \mathbf{Y}, \phi)$, we introduce a generative model in Section A.2 in S1 Appendix that describes the generation of variant read counts $\mathbf{A}$ and the binned read count embedding $\tilde{\mathbf{R}}$. This model (Fig A in S1 Appendix) requires two hyperparameters $c$ and $\alpha$, where $c \in \mathbb{N}$ is the upper bound on the total number of chromosomal copies at any locus in the genome and $\alpha \in [0, 1]$ is the probability of misreading a single nucleotide during sequencing. Importantly, while Definitions 1 to 3 explicitly indicate the number $k$ of clones, the number $k$ of clones is not a hyperparameter and will be part of the inference. Specifically, our generative model enables us to approximate the posterior

probability $P(T, \mathbf{Y}, \phi \mid \mathbf{A}, \mathbf{D}, \tilde{\mathbf{R}}, c, \alpha)$ for a clonal tree $T$ with any number $k$ of nodes and associated clonal genotypes $\mathbf{Y}$ and cell clustering $\phi$ (derivation in Section A.2 in S1 Appendix). However, due to limitations of the sequencing technology the number of clones that are detectable from the data may be fewer than the number of nodes in clonal tree $T$. The ability to detect clone $j$ from the observed data is a function of the sequencing coverage, the number of cells in clone $j$ and the number of SNVs newly introduced in clone $j$. To prevent overfitting of the data, the pre-specified detection threshold $t \in \mathbb{N}$ controls the minimum amount of observed data in support of each inferred clone. See Section A.3 in S1 Appendix for a formal definition and additional details.

This leads to the following problem.

**Problem 1 (Clonal Tree Inference (CTI))** *Given variant reads* $\mathbf{A} \in \mathbb{N}^{n \times m}$, *total reads* $\mathbf{D} \in \mathbb{N}^{n \times m}$, *binned read count embedding* $\tilde{\mathbf{R}} \in \mathbb{R}^{n \times \ell}$, *maximum copy number* $c \in \mathbb{N}$, *sequencing error probability* $\alpha \in [0, 1]$ *and detection threshold* $t \in \mathbb{N}$, *find a clonal tree* $T$ *with detectable clonal genotypes* $\mathbf{Y}$ *and cell clustering* $\phi$ *with maximum posterior probability* $P(T, \mathbf{Y}, \phi \mid \mathbf{A}, \mathbf{D}, \tilde{\mathbf{R}}, c, \alpha)$.

## PHERTILIZER

To solve the CTI problem, PHERTILIZER maintains a set $\mathcal{T}$ of candidate trees throughout three phases: (i) initialization, (ii) growing, and (iii) ranking each tree in $\mathcal{T}$ by its posterior probability. First, in the initialization phase, the set $\mathcal{T}$ is initialized with a single tree containing only a root node $v_1$. All $n$ cells are assigned to the node's cell cluster $\phi(v_1)$ and all genotypes are initialized to $y_{v_1,q} = 1$ for each SNV $q$.

Second, in the growing phase (Section A.4 in S1 Appendix), PHERTILIZER recursively constructs the candidate set $\mathcal{T}$ of clonal trees and the respective clonal genotypes and cell clusterings by performing three different elementary tree operations (Linear, Branching and Identity) on each leaf node $v_j$ of each candidate tree $T \in \mathcal{T}$ (Fig 2). Specifically, each operation takes as input $(T, \mathbf{Y}, \phi)$ and yields a new clonal tree $T'$ with updated genotypes $\mathbf{Y}'$ and cell clustering $\phi'$ by extending leaf $v_j$ of $T$ (Fig 2). The key idea is that each elementary tree

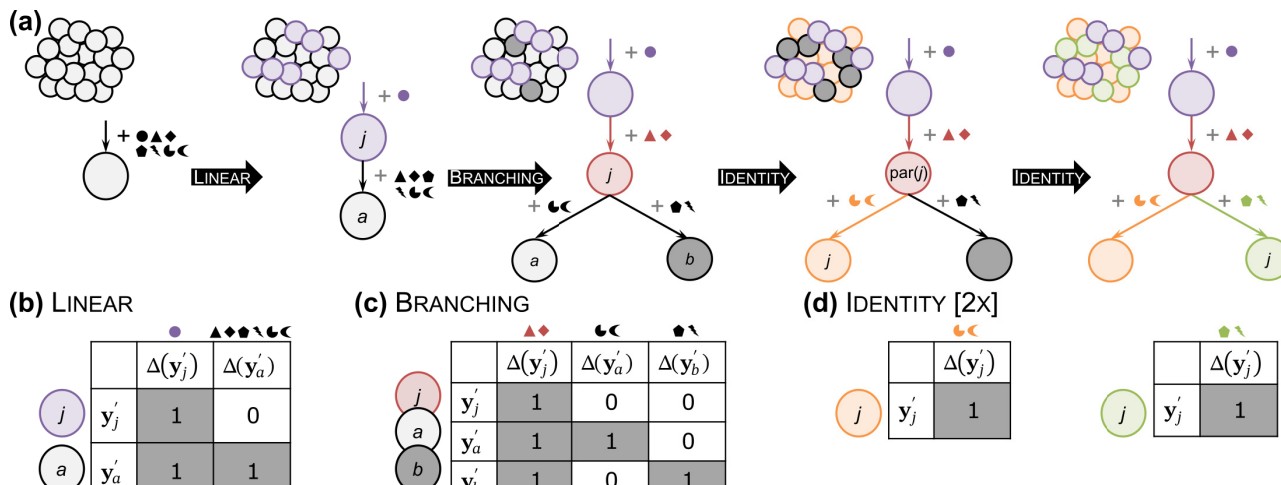

**Fig 2. PHERTILIZER solves the CTI problem by enumerating clonal trees using three elementary operations in a recursive fashion.** (a) Each operation yields a new clonal tree $T'$ by extending a leaf $v_j$ of the previous clonal tree $T$ and reassigning its SNVs $\Delta(\mathbf{y}_j)$ and cells $\phi(j)$. The resulting clonal genotypes $\mathbf{Y}'$ and cell clustering $\phi'$ are constrained as depicted: (b) Linear, (c) Branching, and (d) Identity.

operation breaks down the CTI problem into smaller subproblems. Intuitively, a `Linear` operation (Fig 2a) replaces a leaf node with a two node linear subtree, while a `Branching` operation (Fig 2b) replaces the leaf node with a three node binary subtree. The former represents stepwise acquisition of SNVs with evidence of intermediary clones present in the observed data, while the latter indicates evidence of divergence from a common ancestor [31]. The `Identity` (Fig 2c) operation does not modify the tree and is useful during the growing process where all candidate trees are enumerated.

These operations are defined more formally in Section A.4 in S1 Appendix. While the specifics vary slightly, both `Linear` and `Branching` are solved using a coordinate descent approach. That is, we fix clonal genotypes $\mathbf{Y}'$ and solve for the cell clustering $\phi'$ and alternate. Drawing parallels between image segmentation, where combining both pixel and pixel location features results in better clustering [32], we incorporate the binned read count embedding $\tilde{\mathbf{R}}$ and variant read counts $\mathbf{A}$ into a single feature [11]. We then use this feature as input to the normalized cut algorithm [32] (worst case running time $O(n^3)$) to obtain a cell clustering with two clusters. The advantage of combining SNV and CNA signal into a single feature is that cell clustering is improved when one or both of these signals are weak, subsequently improving the SNV partition which we solve for next. Given a fixed cell clustering $\phi'$, we use our generative model to update clonal genotypes $\mathbf{Y}'$ by assigning each SNV to the node in extended tree $T'$ with maximum posterior probability. This is done in time $O(nm)$. We terminate this process upon convergence or when a maximum number of iterations is reached. This results in the running time of one elementary tree operation as $O(n^3 + nm)$. The resulting clonal tree $T'$ is then appended to the candidate set $\mathcal{T}$ provided all its clone are detectable for a specified detection threshold $t$ and meet additional regularization criteria (Section A.4 in S1 Appendix).

Third, once no new clonal trees are added to the candidate set $\mathcal{T}$, we return the clonal tree $T$, clonal genotypes $\mathbf{Y}$ and cell clustering $\phi$ with maximum posterior probability after post-processing (Section A.4 in S1 Appendix). Importantly, the top down approach by PHERTILIZER requires that no assumptions be made *a priori* regarding the number of nodes $k$ in the inferred clonal tree. PHERTILIZER is implemented in Python 3, open source (BSD-3-Clause), and available at https://github.com/elkebir-group/phertilizer.

## Results

### Simulation study

**Overview.**   To assess the performance of PHERTILIZER and compare it with previously proposed methods, we performed a simulation study with known ground truth clonal trees, evaluating the following four questions:

(i) How accurate are the inferred clonal trees? (ii) How well is each method able to identify clusters of cells with similar clonal genotypes? (iii) How accurate are the inferred clonal genotypes? (iv) How sensitive is each method to violations of the infinite sites assumption? We designed our simulation study to match the characteristics of current datasets generated from ultra-low coverage scDNA-seq. To achieve this, we generated simulation instances with varying number $k \in \{5, 9\}$ of nodes, number $n \in \{1000, 2000\}$ of sequenced cells and number $m \in \{5000, 10000, 15000\}$ of SNVs with a mean sequencing coverage $g \in \{0.01\times, 0.05\times, 0.1\times\}$. We replicated each of these combinations 10 times for a total of 360 instances. See Section B.1 in S1 Appendix for details on the simulation instances.

We assessed the quality of an inferred solution $(T, \phi, \mathbf{Y})$ against a ground-truth tree $T^*$, cell clustering $\phi^*$ and clonal genotypes $\mathbf{Y}^*$ using ancestral pair recall (APR), incomparable pair recall (IPR), and clustered pair recall (CPR) metrics for cells and SNVs [33], as well as genotype similarity. In addition, we computed a single *accuracy* value (domain: [0, 1]) composed of the

weighted average of APR, IPR and CPR—where the weights are proportional to the number of pairs in each class—such that an SNV and cell accuracy of 1 imply that the inferred solution perfectly matches ground truth. The *genotype similarity* equals 1 minus the normalized Hamming distance between ground truth genotypes and inferred genotypes of cells. We refer to Fig 3a Section B.1.4 in S1 Appendix for examples and more formal definitions.

We benchmarked against SBMClone [28] because it is the only existing clustering/genotyping method that co-clusters cells and SNVs. We also benchmarked against a commonly adopted *ad hoc* practice which we refer to as Baseline [6, 7]. In Baseline, cells are first clustered into clones from the read count embedding $\tilde{\mathbf{R}}$. Clonal genotypes are obtained by pooling the reads of all cells assigned to clone $j$ and setting $y_{jq} = 1$ when the ratio of alternate to total reads at each locus $q$ exceeds a threshold of 0.05 (details provided in Section B.1.1 in S1 Appendix). Since there exists no standalone method that performs cell clustering, genotyping and tree inference for ultra-low coverage scDNA-seq, we paired SBMClone and Baseline with SCITE [12]. We attempted to include SCARLET [27] and BiTSC2 [24] in our benchmarking as these scDNA-seq methods incorporate copy number aberrations in their models. However both failed to run on our simulated data. Specifically, SCARLET, which was designed for medium-to-high coverage data, was unable to appropriately handle missing entries in the total read count matrix $\mathbf{D}$. BiTSC2, an MCMC method, was unable to scale to the size of our input datasets—the largest instances considered in the BiTSC2 paper [24] had $n = 500$ cells and $m = 200$ SNVs, significantly smaller than our smallest simulation instances with $n = 1000$ cells and $m = 5000$ SNVs.

For brevity, we focus our discussion on a coverage $g = 0.05\times$ and show aggregated results for $n \in \{1000, 2000\}$ cells and $k \in \{5, 9\}$ clones. We report the median for all performance

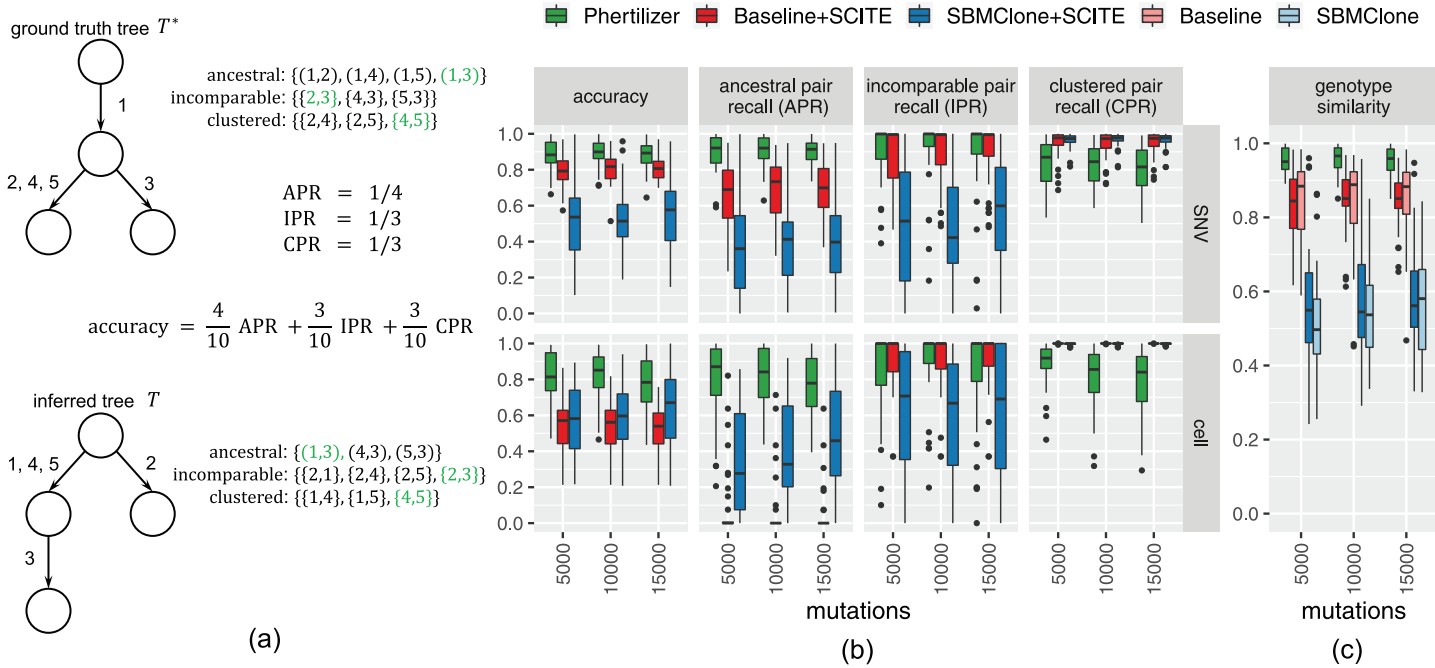

**Fig 3. Phertilizer outperforms Baseline and SBMClone on simulated data.** We show aggregated results for $n \in \{1000, 2000\}$ cells, $k \in \{5, 9\}$ clones and coverage $g = 0.05\times$. (a) Example of SNV accuracy, ancestral pair recall (APR), clustered pair recall (CPR), and incomparable pair recall (IPR) metrics. See Section B.1.4 in S1 Appendix for the corresponding example for cell metrics and genotype similarity. (b) Phertilizer outperforms Baseline +SCITE and SBMClone +SCITE in APR and IPR for both SNVs and cells. Although competing methods rank higher in CPR, overall Phertilizer performs the best considering the accuracy. (c) Phertilizer more accurately recovers clonal genotypes than competing methods.

metrics and include the interquartile range (IQR), i.e., the difference between the 75th and 25th percentiles of the data, when appropriate. We note deviations in trends where relevant, and refer to Section B.1.5 in S1 Appendix for remaining results.

**Results.** We first evaluated the accuracy of SNV placement on the inferred tree by assessing APR, CPR, IPR and accuracy for SNVs (Fig 3b, Fig I in S1 Appendix). Overall, PHERTILIZER achieved the highest SNV accuracy (median: 0.90) in terms of SNV placement of all three methods (BASELINE+SCITE median: 0.81, SBMCLONE+SCITE median: 0.54).

For SNV APR, PHERTILIZER (median: 0.92) outperformed both BASELINE+SCITE (median: 0.71) and SBMCLONE+SCITE (median: 0.38). This implies that our `Linear` operation reliably partitions SNVs accurately and that our `Branching` operation performs well at identifying SNVs that should be placed at the parent node. Moving on to SNV IPR, PHERTILIZER (median: 1.0, IQR: 0.11) outperformed SBMCLONE+SCITE (median: 0.48, IQR: 0.5) with lower variability than BASELINE+SCITE (median: 1.0, IQR: 0.18). Thus, in addition to correctly identifying the SNVs in the parent node, the `Branching` operation also successfully partitions the SNVs in the children nodes. However, for SNV CPR, PHERTILIZER performed worse compared to SBMCLONE+SCITE (median: 0.98) and BASELINE+SCITE (median: 0.97) but still maintained good performance (median 0.84). It is important to note that an SNV CPR of 1 is achievable by clustering all SNVs into a single cluster. Thus, the cost of underfitting the data, or grouping the SNVs into a few very large clusters, will be reflected in decreased APR and IPR. Indeed, we observed this to be the case for both BASELINE+SCITE, which performed relatively poorly on APR, and SBMCLONE+SCITE which was the worst performing on both APR and IPR.

Next, we evaluated the accuracy of the cell clustering and placement on the inferred tree. We observed similar trends across accuracy, APR, IPR and CPR cell performance metrics to those identified for their SNV counterparts (Fig 3b, Fig I in S1 Appendix). PHERTILIZER achieved the highest overall cell accuracy (median: 0.82) in comparison to SBMCLONE+SCITE (median: 0.60) and BASELINE+SCITE (median: 0.56). Likewise, PHERTILIZER (median: 0.84) outperformed all other methods on cell APR (BASELINE+SCITE median: 0.0, SBMCLONE+SCITE median: 0.32). On cell IPR, both PHERTILIZER (median: 1.0, IQR: 0.2) and BASELINE+SCITE (median: 1.0, IQR: 0.13) significantly outperformed SBMCLONE+SCITE (median: 0.67, IQR: 0.68) but BASELINE+SCITE had slightly lower variability (IQR) than PHERTILIZER. Similarly to SNV CPR, BASELINE+SCITE (median: 1.0) and SBMCLONE+SCITE (median: 1.0) outperformed PHERTILIZER on cell CPR (median: 0.87), but the corresponding decreased performance in cell APR and IPR was indicative of inferring too few cell clusters.

In addition to providing more supporting evidence for the validity of our elementary operations, these cell placement performance metrics also highlight the advantage of PHERTILIZER utilizing both copy number and SNV signal for tree inference. In contrast, BASELINE+SCITE prioritizes copy number signal and is unable to further refine a cluster of cells with distinct clonal genotypes but the same copy number profile. Conversely, SBMCLONE+SCITE ignores copy number signal and struggles to infer clones with sparse SNV signal.

Finally, we assessed the genotype similarity and included SBMCLONE and BASELINE as this is obtained prior to tree inference. Since genotype similarity compares the inferred genotype or each simulated cell against its ground truth genotype, it captures the interplay between cell clustering and clonal genotyping. Given that PHERTILIZER achieved the highest performance on the clonal tree inference and cell clustering metrics, we would expect PHERTILIZER to have the highest genotype similarity. Indeed, Fig 3c demonstrates that to be true since PHERTILIZER was the only method to have a median genotype similarity above 0.95. BASELINE was the next highest (median: 0.88), closely followed by BASELINE+SCITE (median: 0.85), while SBMCLONE+SCITE (median: 0.55) and SBMCLONE (median: 0.53) had the worst performance. When evaluating these metrics at the lowest coverage $g = 0.01\times$, PHERTILIZER maintained top performance on

both cell placement and genotype similarity but BASELINE+SCITE was competitive with PHERTI-LIZER on SNV placement (Fig H in S1 Appendix). For the highest coverage $g = 0.1\times$, PHERTILIZER achieved the highest accuracy on SNV placement and cell placement and had a median geno-type similarity of 0.97 while the next closest competitor (BASELINE) had a median similarity of 0.89 (Fig J in S1 Appendix). In terms of running time for the case of $n = 1000$ cells, $m = 15000$ SNVs, and a coverage of $g = 0.01\times$, the median running time of PHERTILIZER was 460 s, 45.9 s for BASELINE+SCITE and 101 s for SBMCLONE+SCITE (Fig M in S1 Appendix).

To perform sensitivity analysis, we generated two additional sets of simulations. The first had the same parameters as above but excluded CNAs, such that every locus was heterozygous diploid. We also excluded BASELINE+SCITE from comparison as only a single clone was inferred after cell clustering. We found PHERTILIZER still outperformed SBMCLONE+SCITE but that for coverage $g = 0.01\times$ our performance were slightly worse than simulations with CNAs (Fig K in S1 Appendix), implying CNA features aid inference when sequencing coverage is extremely sparse. For the second, simulations were generated under a Dollo [34] evolutionary model with $k = 9$, $m = 15000$, coverage $g \in \{0.01\times, 0.05\times, 0.01\times\}$. We found that PHERTILIZER still outperformed BASELINE+SCITE and SBMCLONE+SCITE, having maintained high scores on all performance metrics (Fig L in S1 Appendix) with the exception of cell APR for the lowest coverage.

**Hyperparameter selection.**   PHERTILIZER requires a number of hyperparameters for infer-ence. To assess sensitivity to each of these hyperparameters, we performed an additional simu-lation study varying the base sequencing error probability $\alpha \in \{0.001^*, 0.01\}$, the maximum number of copies $c \in \{3, 5^*, 9\}$, the detectability threshold $t \in \{3^*, 5, 7, 11\}$ in addition to run-time parameters such as the number of restarts ($\{5, 15^*, 30, 60\}$, maximum number of itera-tions ($\{25, 50^*, 100, 200\}$) per elementary tree operation and the quality check upper bound qc $\in \{0.025^*, 0.05\}$ for cells not harboring the specified set of SNVs—see Section A.4.6 in S1 Appendix for details on the quality check. Hyperparameter choices with $*$ indicate the default PHERTILIZER values used in the above simulation study. In all simulated instances, we fixed the number of cells at $n = 1000$, SNVs at $m = 15000$ and sequencing coverage at $g = 0.05\times$— see Fig N and Fig O in S1 Appendix for these results.

Overall, we found that overestimating the base sequencing error $\alpha$ led to a small drop in performance in terms of genotype similarity and SNV and cell accuracy. For the remaining comparisons, we fixed $\alpha = 0.001$ and observed that increasing the maximum copies $c$ to 9 slightly improved overall performance. This is likely attributed to the greater number of allele-specific copy number states that are considered for $c = 9$ than $c = 5$. We also found that perfor-mance remained similar for the detectability threshold until increased to $t = 11$. Due to the low-sequencing coverage, having such a high detectability threshold yields poorly resolved trees with fewer clones than ground truth. However, variation in the quality check threshold did not lead to significant differences in performance. Lastly, we found that 15 restarts with 50 maximum iterations per elementary tree operations was sufficient to maximize performance. However, it may be prudent to increase these values on real data at the cost of increasing run-time. Although this sensitivity analysis provides guidance for the setting of hyperparameter values, we recommend to perform a grid search over hyperparameters, such as $\alpha$ and $c$, when these values are difficult to estimate in practice. The posterior probability approximated by PHERTILIZER can be utilized to discriminate between output clonal trees with different parame-ter settings.

In summary, we conclude that given the high level of accuracy obtained by PHERTILIZER on these performance metrics, not only are the elementary operations successful in isolation but also the posterior probability is useful in discriminating between candidate clonal trees. Addi-tionally, we find that utilizing copy number signal, whenever available, is necessary when

working with ultra-low coverage scDNA-seq data but not sufficient for accurate clonal tree reconstruction and/or SNV genotyping.

## High-grade serous ovarian cancer patient

Utilizing a grid search over input parameters(Section B.2.1 in S1 Appendix), we ran PHERTILIZER on $n$ = 890 DLP+ sequenced cells from three clonally-related cancer cells lines sourced from the same high-grade serous ovarian cancer patient [6]. We used the variant and total read counts **A**, **D** for $m$ = 14, 068 SNVs and derived binned read counts **R** for $b$ = 6, 207 bins (bin width of 500 KB) from data reported by Laks et al. [6]. The average sequencing coverage for these data was 0.25×. Utilizing an approach similar to the BASELINE method described above, Laks et al. [6] identified 9 copy number clones (labeled *A-I*) via dimensionality reduction and density-based clustering and reconciled them in a phylogeny with the copy number clones as leaves (Fig 4a). We annotated inferred trees with mutations in cancer-related genes in the cBioPortal [35, 36] and Cancer Gene Census (CGC) [37] from COSMIC v97 (see Section B.2.2 in S1 Appendix).

As shown in Fig 4b, PHERTILIZER inferred a clonal tree with 13 nodes and 8 clones. We found that PHERTILIZER's tree closely aligned with the Laks et al. [6] tree, with both approaches correctly identifying three major clades corresponding to the three distinct cell lines of origin (Fig

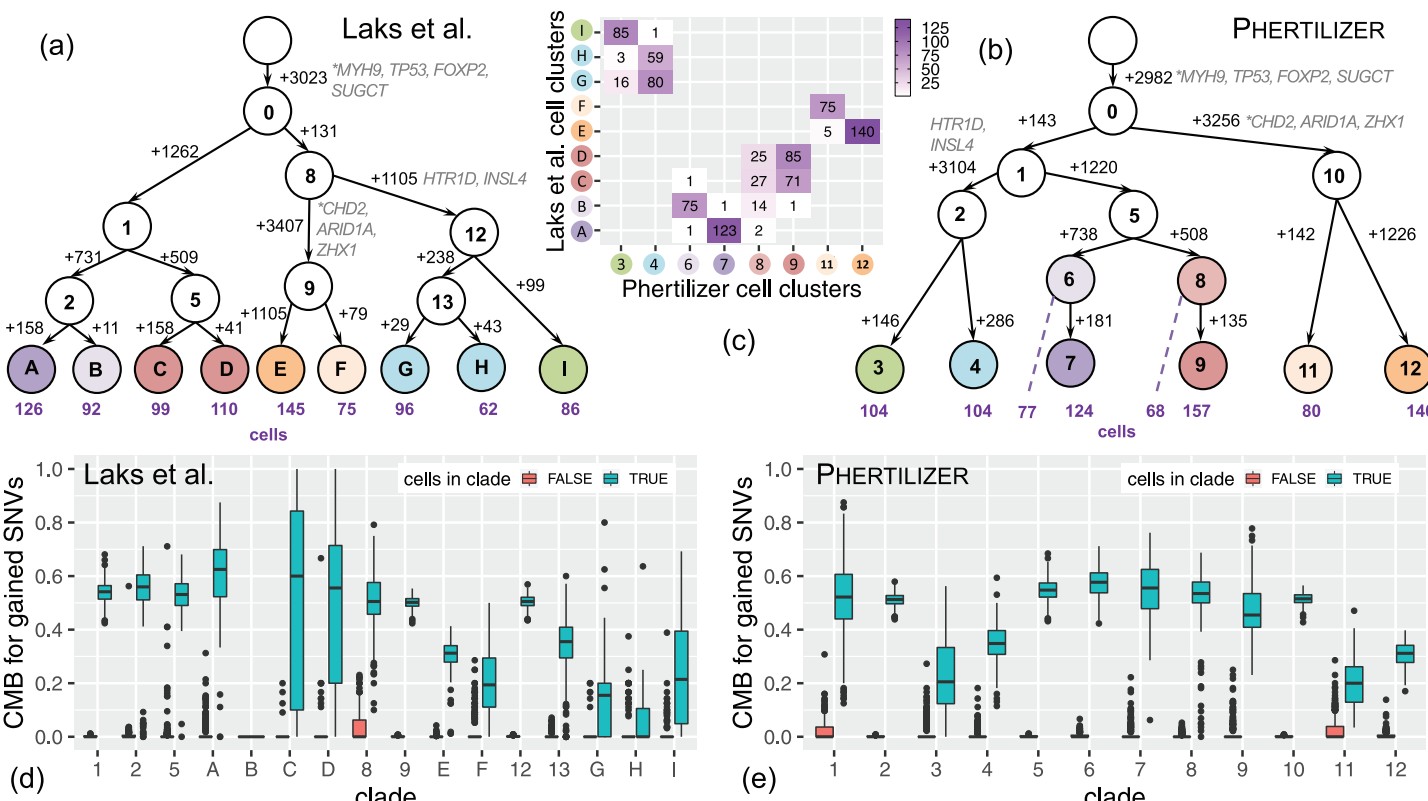

**Fig 4. PHERTILIZER improves upon the SNV phylogeny previously inferred by Laks et al. [6].** (a) The clonal tree inferred by Laks et al. [6] with edges labeled by SNV gains and cell numbers shown below leaf nodes. (b) The clonal tree inferred by PHERTILIZER with edges labeled by SNV gains and cell numbers shown below leaf nodes. Cancer-related genes are labeled next to the SNVs in (a) and (b) with '*' indicating a stop-gain variant. (c) Mapping between the Laks et al. [6] cell placement and PHERTILIZER cell placement. (d-e) Cell mutational burden (CMB) comparison between cells within (blue) and outside (red) of each clade in the inferred Laks et al. [6] and PHERTILIZER clonal tree.

4c). Additionally, driver genes *TP53*, *SUGCT*, and *MYH9* are identified as clonal in both the Laks et.al [6] inferred tree (Fig 4a) and the PHERTILIZER clonal tree (Fig 4b). Similarly, subclonal SNVs in *CHD2*, *ARID1A*, *ZHX1*, *HTR1D*, and *INSL4* were placed in the corresponding clades of both trees. To further assess the quality of each inferred clade, we developed a performance metric called *cell mutational burden* (CMB) defined as $\text{CMB}(i, M) = \sum_{q \in M} \mathbf{1}\{a_{iq} > 0\}/\sum_{q \in M} \mathbf{1}\{d_{iq} > 0\}$.

In words, CMB($i$, $M$) is the fraction of mapped SNV loci $M$ with mapped variant reads in cell $i$. For a specified *clade j* or subtree rooted at node $v_j$, SNVs $M_j$ are the SNVs gained at node $v_j$. For a cell $i$ placed within clade $j$, we expect CMB($i$, $M_j$) to be high, although the value will depend on copy number. By contrast, for cells placed outside of clade $j$, we expect CMB($i$, $M_j$) to be low. More details on CMB are provided in Section B.2.3 in S1 Appendix.

Fig 4d and 4e depicts the comparison of the distributions of CMB for all clades for cells placed within and outside that clade for Laks et al. [6] and PHERTILIZER, respectively. For cells placed outside of a specified clade, the reported tree by Laks et al. [6] as well as the tree inferred by PHERTILIZER have a median CMB of 0 for all clades. This is indicative of high specificity of both methods, i.e., SNVs are not assigned to a clade if there are observations of that SNV outside that clade. However, for cells placed within the clades, we observed greater variability for the Laks et al. [6] inferred clades than PHERTILIZER. This variability was most pronounced for the leaf nodes, especially *C*, *D*, *G*, *F*, *H* and *I*, where most have a small number of SNVs gained. We further analyzed clusters *G* and *H*, where the 25th percentile of the CMB for cells within the clade equals 0, and clusters *C* and *D* that had large variability (IQR of 0.74 and 0.51, respectively).

The location of clusters *G* and *H* in the embedding space suggests overfitting during density-based clustering, with an arbitrary split of a larger cohesive cluster containing *G* and *H* (Fig P in S1 Appendix). In comparison, PHERTILIZER uses both copy number and SNV signal, resulting in the Laks et al. [6]*G* and *H* cells being clustered together to node 4 in the inferred PHERTILIZER clonal tree. Comparing CMB distributions for cells within clades *G* and *H* (Fig 4d) with PHERTILIZER's inferred clade 4 (Fig 4e), we observed a higher 25th percentile (0.31) for node 4 than for *G* (0.00) and *H* (0.00). This resulted in large separation between the CMB distributions of cells within and outside of clade 4 for PHERTILIZER but not for Laks et al. [6]*G* and *H* clades, implying better SNV placement in PHERTILIZER's clonal tree.

The last major difference of note between the two inferred trees is the clustering of the cells in Laks et al. [6] nodes *C* and *D* versus PHERTILIZER's nodes 8 and 9. Similarly to nodes *G* and *H*, we did not observe a clear separation of *C* and *D* in the embedding space (Fig P in S1 Appendix), making it difficult to define clusters of these cells without more rigorous copy number profiling and clustering. However, as we saw in the simulation study, PHERTILIZER was able to detect further SNV evolution within a set of cells having the same copy number profile. Cells in these clusters would not be split in half in the embedding space but instead should be scattered randomly throughout a single cluster in the embedding space. In addition to having observed cells in nodes 8 and 9 randomly scattered in a cluster in the embedding space (Fig P in S1 Appendix), we also observed a clear separation between the CMB distributions for cells placed within and outside of clades 8 and 9 (Fig 4e). For these clades, we also observed low variability with IQRs of 0.08 and 0.13, respectively, whereas clusters *C* and *D* have very high variability in the inferred Laks et al. [6] clonal tree.

Overall, both inferred clonal trees are very similar but since PHERTILIZER simultaneously uses both CNA and SNV information, we obtained a slight improvement in terms of SNV phylogeny inference. Additionally, we note that the small number of SNVs gained at most of the leaf nodes is a direct result of the bottom up approach taken by Laks et al. [6], which performed pseudobulk SNV calling individually on each cell cluster. When SNVs are present in

multiple cell clusters but at low prevalence in each cluster, they may not pass filtering of current somatic SNV callers for a cell cluster. This gives the appearance that SNVs are unique to a single clone, when in actuality they are present in multiple clones but have not been called. When sequencing coverage is closer to the 0.01× as opposed to the 0.25× that we have with these data, correctly genotyping a cell cluster becomes more challenging. In contrast, the top-down approach of PHERTILIZER is better suited to detect SNVs present in multiple cell clusters at low prevalence.

## Eight triple negative breast tumors

We applied PHERTILIZER to eight triple negative breast tumors sequenced via ACT [7], labeled TN1 to TN8. After dimensionality reduction of normalized and GC-bias corrected binned read counts **R**, Minussi et al. [7] identified for each tumor two sets of cell clusters with varying granularity, denoted as superclones and subclones. To obtain the input set of SNVs for each patient, we performed SNV calling of a pseudobulk sample of pooled sequenced cells using MuTect2 in tumor-only mode [38]. See Section B.2.5 in S1 Appendix for further details on data processing. Table A in S1 Appendix displays the breakdown of each tumor in terms of the number $n$ of cells, the number $m$ of SNVs and average coverage $g$, and depicts the number of inferred clones by PHERTILIZER, Minussi et al. [7] superclones and subclones, as well as the number of inferred clones by SBMCLONE. Note that these data have a markedly lower coverage (ranging from 0.017× to 0.039×) than the DLP+ data (0.25×). We additionally ran BASELINE +SCITE with the cell clusters fixed to the Minussi et al. [7] subclones but all instances except TN3 and TN5 timed out after 10 hours. However, the CMB distribution for the inferred trees for TN3 and TN5 (Fig S in S1 Appendix) provides no evidence in support of these trees. SBMCLONE only inferred a single clone for all patients, while PHERTILIZER inferred a tree with more than one clone for 4 out of the 8 tumors (TN1: 6, TN2: 6, TN4: 4 and TN8: 2). These four tumors have the highest average coverage of the eight patients. We will focus our discussion on the clonal trees inferred by PHERTILIZER for tumors TN1 (Fig 5) and TN2 (Fig T in S1 Appendix).

For tumor TN1, PHERTILIZER inferred a branching tree with 11 nodes and 6 clones (Fig 5a). We also identified a subclonal missense SNV in driver gene *DICER1* associated with tumorigenesis and poor prognosis [39, 40]. We noted good concordance between the Minussi et al. [7] superclones and the PHERTILIZER cell clusters, with the exception of 35 cells that appear as outliers in superclone 1 (Fig 5b, Fig R in S1 Appendix). This suggests these cells might fit better

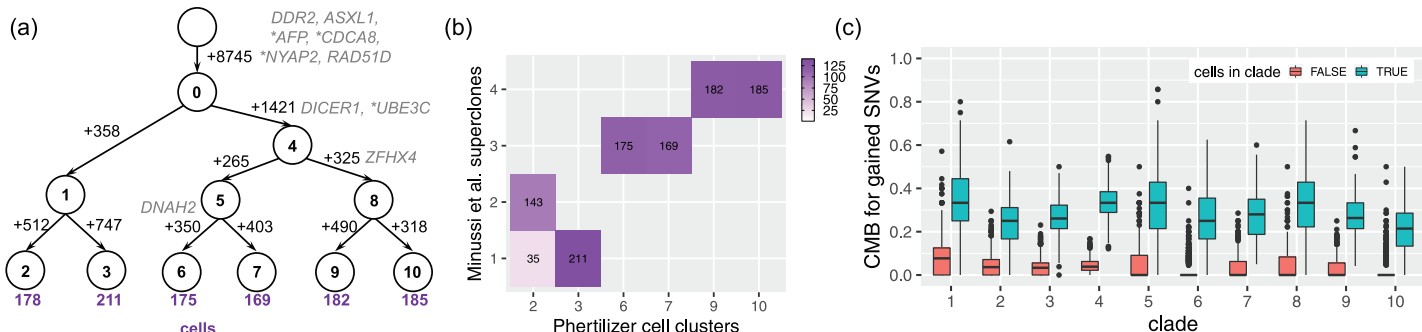

**Fig 5. PHERTILIZER infers clonal tree for breast cancer tumor TN1.** (a) The tree inferred by PHERTILIZER with numbers of SNVs labeled beside the edges, and numbers of cells labeled beneath the leaves. Cancer-related genes are labeled next to the SNVs ('*': stop-gain variant). (b) A mapping between PHERTILIZER's cell clusters and the Minussi et al. [7] superclones. (c) The cell mutational burden (CMB) comparison between cells within (blue) and outside of (red) each clade in the inferred clonal tree.

in superclone 2 based on SNV signal. In addition, we identified 8745 of the 13934 SNVs as truncal. This large truncal distance and branching structure were in alignment with the truncal distance in the clonal lineage tree inferred by Minussi et al. [7] using bulk whole exome sequencing. We used CMB to assess the performance of SNV and cell placement (Fig 5c). For clades 5 through 10, we observed that the median CMB for cells outside of the clade is 0. Clades 9 and 10 were particularly interesting because the embedding space depicts the occurrence of SNV evolution within Minussi et. al.'s [7] superclone 4 (Fig R in S1 Appendix). For clades 2 through 4, we noted the median CMB for cells outside of the clade was around 0.035 for each of these clades, while clade 1 is the highest at 0.077. Upon further investigation of these 358 SNVs, we found that the median number of mapped reads was 5 when aggregating all cells, making these SNVs especially challenging to place. This drop in performance on the median CMB for cells outside of a clade when compared to the ovarian cancer patient analyzed above is expected due to the drop in sequencing coverage from 0.25× to 0.031×. However, we still observed a large separation between CMB distributions for the cells within the clade and cells outside the clade for all clades.

For tumor TN2, we inferred a branching clonal tree with 11 nodes and 6 clones (Fig T in S1 Appendix). Two of PHERTILIZER's cell clusters directly agreed with Minussi et al. [7] super-clones. However, PHERTILIZER split the remaining two superclones into four cell clusters (3, 4, 5, 6) using SNV information. We observed low median CMB (0) for cells outside of the clade and a distinct separation between the cells within and outside of the clade distributions, providing evidence for this cell clustering and SNV placement. For tumor TN4, we identified an 8-node tree with five cell clusters, with trends similar to tumors TN1 and TN2 in terms of cell clustering concordance and CMB (Fig U in S1 Appendix). Finally, for tumor TN8 at the lowest coverage 0.021× of these four tumors, PHERTILIZER only inferred a 3-node branching tree with two cell clusters (Fig V in S1 Appendix).

## Discussion

Ultra-low coverage scDNA-seq has greatly enhanced our ability to study tumor evolution from a copy number perspective [7, 17]. Capitalizing on the strong copy number signal inherent in this data, we proposed a new method, PHERTILIZER, that grows an SNV phylogeny in a recursive fashion using elementary tree operations. We demonstrated the effectiveness of our approach relative to existing clustering methods on both simulated and real data. Importantly, we found that for the current number of cells (800 − 2000) used in practice, PHERTILIZER performs markedly better than these methods, yielding more accurate clonal trees, cell clusters, and clonal genotypes. As the first method to reconstruct the evolutionary history of SNVs from ultra-low coverage scDNA-seq, PHERTILIZER helps advance the study of tumor evolution and makes progress towards the goal of joint SNV and CNA phylogeny inference at single-cell resolution.

There are several additional limitations and directions for future research. First, as sequencing coverage drops below 0.02× as in the ACT data, PHERTILIZER does not infer clonal trees with more than one clone. Although inference is impacted by numerous factors, like copy number profile, it does perhaps suggest a coverage of approximately 0.02× as the limit of detection for PHERTILIZER. Second, accurate variant calling also remains an open problem for ultra-low scDNA-seq data, making it challenging to identify input subclonal variants. Beyond scDNA-seq data, accurate SNV variant calling from scRNA-seq and ATAC-seq datasets is also challenging [41, 42]. But new methods, such as Monopogen [43], SComatic [44], VarCA [45], scAllele [46], reference-based methods [47] and the use of patient derived cell lines [48] are rapidly improving our ability to accurately conduct single-cell somatic mutational profiling

from diverse technologies beyond scDNA-seq. By accounting for different error profiles in variant calling, PHERTILIZER could be extended to directly model the evolution of somatic variants from scRNA-seq and ATAC-seq datasets. Third, beyond SNVs and CNAs we plan to support structural variants and integrate other omics modalities such as methylation and transcription [49]. Fourth, the infinite sites model used in this work is often violated due to copy-number deletions. While we demonstrated robustness to such violations, a future direction is to use the Dollo evolutionary model [14, 34]. Finally, our model lacks an explicit placement of CNA events on the tree. Tree reconciliation methods, such as PACTION [50], can now be applied to integrate an SNV clonal tree generated by PHERTILIZER and a CNA tree to obtain a joint tree.

## Supporting information

**S1 Appendix. Supplementary materials.**
(PDF)

## Acknowledgments

We thank the Navin Lab and Darlan Minussi for their assistance with the ACT data. Additionally, we thank the Shah Lab, including Daniel Lai, Robert Reinert, and Andrew McPherson, for their assistance with the DLP+ data.

## Author Contributions

**Conceptualization:** Idoia Ochoa, Mohammed El-Kebir.

**Data curation:** Leah L. Weber, Chuanyi Zhang.

**Formal analysis:** Leah L. Weber, Chuanyi Zhang, Idoia Ochoa, Mohammed El-Kebir.

**Funding acquisition:** Mohammed El-Kebir.

**Investigation:** Leah L. Weber, Chuanyi Zhang, Idoia Ochoa, Mohammed El-Kebir.

**Methodology:** Leah L. Weber, Chuanyi Zhang.

**Project administration:** Idoia Ochoa, Mohammed El-Kebir.

**Resources:** Leah L. Weber, Chuanyi Zhang.

**Software:** Leah L. Weber, Chuanyi Zhang.

**Supervision:** Idoia Ochoa, Mohammed El-Kebir.

**Validation:** Leah L. Weber, Chuanyi Zhang.

**Visualization:** Leah L. Weber.

**Writing – original draft:** Leah L. Weber, Chuanyi Zhang, Idoia Ochoa, Mohammed El-Kebir.

**Writing – review & editing:** Leah L. Weber, Chuanyi Zhang, Idoia Ochoa, Mohammed El-Kebir.

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
