## [Decision Letter · Decision Letter 0]

23 Aug 2023

Dear Dr. El-Kebir,

Thank you very much for submitting your manuscript "Phertilizer: Growing a clonal tree from ultra-low coverage single-cell DNA sequencing of tumors" for consideration at PLOS Computational Biology. As with all papers reviewed by the journal, your manuscript was reviewed by members of the editorial board and by several independent reviewers. The reviewers appreciated the attention to an important topic. Based on the reviews, we are likely to accept this manuscript for publication, providing that you modify the manuscript according to the review recommendations.

Sincerely,

Teresa M. Przytycka

Academic Editor

PLOS Computational Biology

Feilim Mac Gabhann

Editor-in-Chief

PLOS Computational Biology

Reviewer's Responses to Questions

**Comments to the Authors:**

Reviewer #1: The authors addressed all concerns in a proper way and, as a result, the paper results more solid.

Overall, the article is methodologically sound and provides an interesting contribution to the field of cancer evolution inference from single-cell data.

My only minor concern is the lack of a proper discussion on the strategies for calling SNVs from single-cell RNA-seq or ATAC-seq data, the availability of which is rapidly growing, particularly thanks to the increasing number of experiments on patient-derived models. See, for instance (but not only):

- Muyas, F., Sauer, C.M., Valle-Inclán, J.E. et al. De novo detection of somatic mutations in high-throughput single-cell profiling data sets. Nat Biotechnol (2023). https://doi.org/10.1038/s41587-023-01863-z

- Ramazzotti, D., Angaroni, F., Maspero, D. et al. Variant calling from scRNA-seq data allows the assessment of cellular identity in patient-derived cell lines. Nat Commun 13, 2718 (2022). https://doi.org/10.1038/s41467-022-30230-w

- Arya R Massarat and others, Discovering single nucleotide variants and indels from bulk and single-cell ATAC-seq, Nucleic Acids Research, Volume 49, Issue 14, 20 August 2021, Pages 7986–7994, https://doi.org/10.1093/nar/gkab621

- Patricia M Schnepp and others, SNV identification from single-cell RNA sequencing data, Human Molecular Genetics, Volume 28, Issue 21, 1 November 2019, Pages 3569–3583, https://doi.org/10.1093/hmg/ddz207

- Liu, F., Zhang, Y., Zhang, L. et al. Systematic comparative analysis of single-nucleotide variant detection methods from single-cell RNA sequencing data. Genome Biol 20, 242 (2019). https://doi.org/10.1186/s13059-019-1863-4

The exploitation of single-cell mutational profiles generated from such data types might allow for the reconstruction of reliable models of cancer evolution, and may represent both an alternative and a complement (in case of multi-modal assays) to scDNA-seq data.

Reviewer #2: All the issues pointed out in the RECOMB version have been addressed. I have no other requests.

**Have the authors made all data and (if applicable) computational code underlying the findings in their manuscript fully available?**

Reviewer #1: Yes

Reviewer #2: Yes

PLOS authors have the option to publish the peer review history of their article (what does this mean?). If published, this will include your full peer review and any attached files.

Reviewer #1: No

Reviewer #2: No

Figure Files:

Data Requirements:

Reproducibility:

References:

---

## [Editor Report · Decision Letter 1]

26 Sep 2023

Dear Dr. El-Kebir,

We are pleased to inform you that your manuscript 'Phertilizer: Growing a clonal tree from ultra-low coverage single-cell DNA sequencing of tumors' has been provisionally accepted for publication in PLOS Computational Biology.

Best regards,

Teresa M. Przytycka

Academic Editor

PLOS Computational Biology

Feilim Mac Gabhann

Editor-in-Chief

PLOS Computational Biology

---

## [Editor Report · Acceptance letter]

2 Oct 2023

PCOMPBIOL-D-23-01035R1 

Phertilizer: Growing a clonal tree from ultra-low coverage single-cell DNA sequencing of tumors

Dear Dr El-Kebir,

I am pleased to inform you that your manuscript has been formally accepted for publication in PLOS Computational Biology. Your manuscript is now with our production department and you will be notified of the publication date in due course.

With kind regards,

Dorothy Lannert
